# A Novel Non-Invasive Selection Criterion for the Preservation of Primitive Dutch Konik Horses

**DOI:** 10.3390/ani8020021

**Published:** 2018-02-01

**Authors:** Sharon May-Davis, Wendy Y. Brown, Kathleen Shorter, Zefanja Vermeulen, Raquel Butler, Marianne Koekkoek

**Affiliations:** 1Canine and Equine Research Group, University of New England, Armidale, NSW 2351, Australia; wbrown@une.edu.au (W.Y.B.); kshorter@une.edu.au (K.S.); 2Equine Studies, 4271 Dussen, The Netherlands; info@equinestudies.nl; 3Integrated Veterinary Therapeutics, Table Top, NSW 2640, Australia; integratedvettherapeutics@gmail.com; 4JK Equine Balance, 3612 Tienhoven, The Netherlands; jannekekoekkoek@hotmail.com

**Keywords:** Dutch Konik, Metacarpal, Metatarsal, primitive horse, splint bones, Tarpan

## Abstract

The Dutch Konik is valued from a genetic conservation perspective and also for its role in preservation of natural landscapes. The primary management objective for the captive breeding of this primitive horse is to maintain its genetic purity, whilst also maintaining the nature reserves on which they graze. Breeding selection has traditionally been based on phenotypic characteristics consistent with the breed description, and the selection of animals for removal from the breeding program is problematic at times due to high uniformity within the breed, particularly in height at the wither, colour (mouse to grey dun) and presence of primitive markings. With the objective of identifying an additional non-invasive selection criterion with potential uniqueness to the Dutch Konik, this study investigates the anatomic parameters of the distal equine limb, with a specific focus on the relative lengths of the individual splint bones. Post-mortem dissections performed on distal limbs of Dutch Konik (*n* = 47) and modern domesticated horses (*n* = 120) revealed significant differences in relation to the length and symmetry of the 2nd and 4th Metacarpals and Metatarsals. Distal limb characteristics with apparent uniqueness to the Dutch Konik are described which could be an important tool in the selection and preservation of the breed.

## 1. Introduction

Efforts to conserve the uniqueness of primitive horse breeds has led to the establishment of captive breeding programs specifically designed to oversee their breeding selection and re-introduction back into the wild [1]. One such breed is the Konik, a hardy, stocky horse first established in Poland and believed to have descended directly from the now extinct Tarpan (*Equus caballus gmelini*, Antonius) [2,3]. The Tarpan’s disappearance in the wild by 1880, and in- captivity by 1909, saw enthusiasts attempt to “breed-back” the Tarpan by selecting only those horses that bore a striking resemblance [2,3,4]. The selection criterion was based on phenotypic characteristics from the reported sightings of Tarpans dated between 1518–1909 [4,5]. By the mid 1920s, a “re-constructed Tarpan” was established and became known as the Polish Konik. However, these ideals were soon interrupted by World War II and subsequently, the reduced population experienced a genetic bottleneck [2,5,6]. Despite this setback, a breed registry was issued in 1955 and the first volume of the Polish Konik Studbook became established in 1962. This effectively provided the mechanisms to manage selection, breeding and genetic conservation of the Polish Konik horse [2,5,6,7].

The probable genetic contribution from Tarpan ancestry is the plausible logic behind the Polish Konik’s innate ability to survive in harsh environments, while still remaining healthy and fertile [2]. This hardiness was a major factor when considering their selection in the rehabilitation of natural landscapes and sensitive ecosystems in Poland, currently under threat from encroaching forests [8,9]. In a bygone era, the seedlings and saplings were browsed by large herbivores that foraged throughout the region; but in their absence, have been growing unchecked and are now dominating the landscape. Hence, environmentalists and scientists saw the potential of Polish Konik horses for controlling this new invasion of dense forests, whilst still conserving natural wetland areas that were accustomed to ungulate grazing [9,10,11]. Consequently, these large herbivores with natural adaptive instincts have been exported to many European countries with wetland management as a key agenda [8,9].

With this in mind, the selection of Polish Konik horses for The Netherlands proved to be a natural choice when considering the preservation of its extensive wetlands and waterways; and as a large herbivore, these primitive horses could take back control from the advancing forests whilst surviving in the wild with minimal managerial intervention [9,10,11,12]. In addition, the likely genetic link to the extinct Tarpan that once foraged here, guaranteed an innate ability to adapt to these regions and thus ensure its survival. Now, some 33 years later, these primitive horses are referred to as Dutch Koniks and typically require little management [2]. However, the organisations involved in their management are also responsible for maintaining the genetic purity of the breed, in conjunction with maintaining large holdings of land or nature reserves where they range [11,12]. Therefore, in the absence of natural predators and with limited land available in nature reserves, selective culling is sometimes essential to prevent overstocking and potential starvation [13,14].

This selection process aims to maintain the most favourable phenotypic characteristics for the Dutch Konik horse (as stipulated in the Stud-book of Origin of Konik Polski Breed); these include: height at the wither between 130–140 cm (Figure 1), colour (various shades of mouse to grey dun), presence of primitive markings (predominantly a dorsal stripe and leg barring per horse) and no white body hairs [15]. These individual criteria have high heritability [7,16,17,18] and subsequently, the selection process for culling has become problematic due to the high uniformity that now exists within the breed expressing these precise traits. Therefore, additional criteria are desirable to assist in the selection process. This study aims to identify specific anatomic traits that are unique to the Dutch Konik horse to augment the current selection criteria.

The transition to a single-hoofed (monodactyl) species from its multi-toed (polydactyl) ancestors suggests that the distal limb may be an important area of focus when comparing primitive and modern horses. Recent studies of the Dutch Konik’s forefathers in Poland have noted the relative uniformity and variances of certain exterior traits in the distal limb, including the circumference and length of the 3rd Metacarpal (MC3) and 3rd Metatarsal (MT3) [19,20].

Located caudomedial and caudolateral to MC3 and MT3 are 2 lesser Metacarpals and Metatarsals; the 2nd (MC2 and MT2) and 4th (MC4 and MT4) respectively [21]. Collectively, these are often referred to as splint bones and in nearly all horses, there is a nodule located at the distal extremity. These nodules are often visible, quite pronounced and easy to palpate [21]. The position of these nodules provides a simple indication of the relative lengths of the associated Metacarpals (MC2 and MC4) and Metatarsals (MT2 and MT4), remnants of the ancestral toes. Early investigations of the evolution of the equine foot suggested that these splint bones reduced symmetrically [22]. With the view to identify a non-invasive selection criterion with potential uniqueness to the Dutch Konik, this study investigates the anatomic parameters associated with the splint bones of the distal limb in the Dutch Konik in comparison with the modern Domesticate horse, with a specific focus on the relative lengths of the individual splint bones.

## 2. Materials and Methods

### 2.1. Ethical Statement

No horses were euthanized for the purpose of this study and all measurements were obtained post mortem.

### 2.2. Animal Details

Dissections were performed on 47 distal limbs of Dutch Konik horses, and 120 distal limbs from modern domesticated horses. The Dutch Konik horses were sourced from 3 unrelated populations within The Netherlands; 2 females (16 months and 7 years old; maternally related) from de Rug; 2 males (2 and 3 years old; maternally related) from Loevestein; 31 mixed aged and gender legs from Leeuwin (15 forelimbs and 16 hind limbs). The 30 domesticated horses were sourced from 5 countries: United Kingdom (1), New Zealand (2), Japan (4), The Netherlands (7) and Australia (16); and comprised 10 breeds: Thoroughbred (10), Warmblood (4), Australian Stock Horse (3), Crossbred (3), Quarter Horse (2), Welsh Mountain pony (2), Exmoor pony (2), Japanese pony (2), Andalusian (1) and Icelandic (1). The domesticated horses comprised 15 males, 14 females and 1 unknown gender; and at the time of death were aged between stillborn (9.5 month premature) and 30 + years.

### 2.3. Dissections and Measurements

All measurements were performed by the 1st author (an experienced equine anatomist, with a measurement reliability: ±0.7 mm) according to the following procedure using a combined manual and digital Mitutoyo Digimatic Calipers (Mitutoyo Corporation, Kawasaki, Kanagawa Prefecture, Japan) with an associated measurement accuracy of ±0.05 mm for manual and ±0.005 mm for digital measures. Prior to skinning, each limb was palpated for the identification of the distal nodules pertaining to the Metacarpals and Metatarsals. Once skinned, flexor tendons and the 3rd Interosseous muscle (IM3) were removed from the palmer surface of MC3 and the plantar surface of MT3. During this process, it was noted that strong chord-like bands originating from the distal nodules of MC2, MC4, MT2 and MT4 were present in Dutch Konik horses and required resection. To access the distal condyles of MC3 and MT3, the extensor tendons, IM3 branches, collateral ligaments and joint capsules were resected so to disarticulate MC3 and MT3 from the 1st Phalanx (PI).

Nodule to condyle measurements referring to MC2, MC4, MT2, and MT4 are defined in Table 1 (Figure 2) and will be denoted as mMC2, mMC4, mMT2, and mMT4, respectively, and the distances reported in millimeters. The greater the measurement in distance from the nodule to the condyle, the shorter the lesser Metacarpal or lesser Metatarsal in length. Comparisons were also made between the 2nd and 4th Metacarpal (or Metatarsal) of each distal limb, to quantify the degree and direction of asymmetry.

### 2.4. Statistical Analysis

Descriptive analysis was undertaken in Microsoft Excel 2016 (Microsoft Inc., Redmond, WA, USA) to determine the median, mean, standard deviation and range of the nodule to condyle measurements (mMC2, mMC4, mMT2, and mMT4). Inferential statistics between the Dutch Konik and modern domesticated horses were only conducted based on the relative proportion data from each limb to negate the influence of stature. The relative proportion for each limb was defined by the respective difference between the mMC2 (or mMT2) and the mMC4 (or mMT4) and expressed in relation to the mMC4 (or mMT4), whereby a value approaching zero would indicate symmetry. For each limb, as data did not meet assumptions of normality, Mann-Witney U Tests with a significance level of *p* = 0.05 were conducted using SPSS V24 (IBM Statistics, Armonk, NY, USA), with effect size calculated in accordance with Cohen’s d.

## 3. Results

### 3.1. Dutch Konik Horse—Forelegs

In each of the 23 Dutch Konik forelegs measured in this study, the length of MC2 was always greater than that of MC4 (MC2 > MC4) as indicated by the smaller measurements of mMC2 compared to mMC4 (mMC2 < mMC4) shown in Figure 1. The average distances (±SD) from the nodule to the condyle of MC2 (mMC2) in left and right forelegs were 61.2 ± 5.9 mm and 57.7 ± 4.6 mm respectively; whereas the average mMC4 in left and right forelegs were 88.8 ± 19.2 mm and 84.0 ± 18.9 mm. Therefore, MC2 was on average 27.6 ± 19.3 mm longer than MC4 for the left forelegs, and 26.3 ± 18.4 mm for the right.

### 3.2. Dutch Konik Horse—Hindlegs

As for the forelegs, the length of MT2 was always greater than that of MT4 (MT2 > MT4) in each of the 24 Dutch Konik hindlegs measured in this study, corresponding to the smaller measurements of mMT2 compared to mMT4 (mMT2 < mMT4). The average distances (±SD) from the nodule to the condyle of MT2 (mMT2) in left and right hindlegs were 73.5 ± 9.2 mm and 75.8 ± 9.0 mm respectively; whereas the average mMT4 in left and right hindlegs were 106.6 ± 20.2 mm and 110.6 ± 23.3 mm. Therefore, MT2 was on average 33.1 ± 18.8 mm longer than MT4 for the left hindlegs, and 34.8 ± 18.3 mm for the right.

### 3.3. Domesticate Horse—Forelegs

Of the 60 Domesticate horse forelegs measured, the length of MC2 was greater than that of MC4 (MC2 > MC4) in approximately half (*n* = 27) of the forelegs: 13 left and 14 right. The average distances (±SD) from the nodule to the condyle of MC2 (mMC2) in left and right forelegs were 61.9 ± 15.1 mm and 63.0 ± 14.2 mm respectively, whereas the average mMC4 in left and right forelegs were 63.2 ± 13.7 mm and 64.5 ± 13.9 mm. There was on average a much smaller variation between the length of MC2 and MC4 in the Domesticate horse; 5.5 ± 5.2 for the left forelegs, and 5.3 ± 4.9 mm for the right.

### 3.4. Domesticate Horse—Hindlegs

Of the 60 Domesticate horse hindlegs measured in this study, the length of MT2 was greater than that of MT4 (MT2 > MT4) in 34 of the hindlegs: 17 left and 17 right. The average distances (±SD) from the nodule to the condyle of MT2 (mMT2) in left and right hindlegs were 81.6 ± 20.4 mm and 81.9 ± 19.5 mm respectively; whereas the average mMT4 in left and right hindlegs were 85.6 ± 19.0 mm and 84.6 ± 17.6 mm. The average variation between the length of MT2 and MT4 was 8.9 ± 8.7 for the left hindlegs, and 7.1 ± 6.8 mm for the right.

### 3.5. Dutch Konik versus Domesticate

A significant difference in the relative proportions between mMC2:mMC4 and mMT2:mMT4 and moderate effect size were established between the Dutch Konik horse and Domesticate horse for each limb (Left forelimb: U = 330, *p* < 0.001, d = 0.65; right forelimb U = 310, *p* < 0.001, d = 0.67; left hindlimb U = 301, *p* < 0.001, d = 0.63 and right hindlimb U = 370, *p* < 0.001, d = 0.71). Collectively, these results indicate a meaningful significant difference between the Dutch Konik and Domesticate Horse in relation to the symmetry of the 2nd and 4th Metacarpals and Metatarsals, with the Domesticate Horse showing greater symmetry in the splint bones whereas in the Konik, the 2nd Metacarpal and Metatarsal is greater in length than the 4th (Figure 3).

## 4. Discussion

This study, investigating the anatomic parameters associated with the splint bones of the distal equine limb, revealed significant differences between the Dutch Konik and modern Domesticate horse and identified a non-invasive selection criterion with apparent uniqueness to the Dutch Konik.

Consistent with our findings, a 1975 anatomic text describes the 2nd and 4th Metacarpals in the Domesticate horse as variable, but generally equal in length, and located between two-thirds to three-quarters along the length of MC3 [21]. These references to the length of splint bones still exist for the Domesticate horse in scientific literature with the further adage that they are vestigial [23,24]. In contrast to this, the Dutch Konik horses examined in this study showed large variation in length between the 2nd and 4th Metacarpals, with the 2nd always longer than the 4th.

It would appear that, for the modern Domesticate horse, a reduction of the medial and lateral polydactyl digits to splint bones together with the elongation of MC3 and MT3 for speed outweighed the demand for stabilisation or loadbearing [23]. This decreased the energetic cost of locomotion by lessening distal limb mass [23]. By comparison, it seems that increased stabilization and/or loadbearing was of greater advantage to the ancestors of the Dutch Konik. Adapting to a forested landscape with few predators, as opposed to open plains and the presence of cursorial carnivores, offers a plausible explanation. Further investigation into other primitive or related equidae is needed to determine whether this unique skeletal expression is a breed anomaly or a primitive trait. 

In 1935, it was postulated that all eight splint bones reduced symmetrically in 4- and 3-toed prehistoric polydactyl horses based on measurements in 19 specimens including the 4-toed Eohippus, the 3-toed Mesohippus, Merychippus, Hypohippus and Neohipparion; and the monodactyl Equus scotti [22]. The relative symmetry demonstrated in these specimens is reflective of our findings in the Domesticate horse, and at odds with our findings in the Dutch Konik, suggesting a breed anomaly.

Functional studies of the three Metacarpal bones in the equine note that the largest and more significantly loadbearing is MC3, whilst MC2 supports the medial carpals, with the lesser MC4 helping to support the lateral carpals, similarly, this also applies to the Metatarsals [21]. In addition, MC2 and MC4 support MC3 in torsional stress and assist against cantilever bending, whilst decreased bending stresses are noted in the proximal bones. As bending stresses are greater distally, it correlates to the tapering of MC2 and MC4 to the distal nodule [24]. The cited author refers to the latter as the relationship between ‘form and function’, with MC2 and MC4 being geometrically larger proximally and therefore, more supportive functionally. At this point in time, there is no significant literature describing the relevance in breeds of Metacarpal or Metatarsal asymmetry, nor its relevance or potential functional ramification. However, as these loads are measured to the proximal extremities with no distal relevance, this should not affect the function of the reduced distal lengths pertaining to MC4 or MT4 in Dutch Konik horses.

Finally, as this study proposes new criteria in the selection process of Dutch Konik horses, it would be relevant to ascertain heritability. With MC3 and MT3 circumference and length linked to high heritability in Polish Konik horses [8], it would be reasonable to assume similar heritability in the Dutch Konik. The question is whether it applies to MC2, MC4 and MT2, MT4? With few founding ancestors [2,3,4] and samples derived from three differing geographical populations, it could be postulated that the lengths of MC2, MC4 and MT2, MT4 are indeed a heritable trait due to Cohen’s d calculations describing the mean variances as highly repeatable within the breed. Furthermore, the significant asymmetrical variances shown in this study between the measurements of MC2, MC4 and MT2, MT4 in Dutch Konik horses, do not concur with previous studies or anatomic text, reporting symmetrical reduction or morphology in Metacarpals and Metatarsals [21,22,24]. Nor do they correspond to the Domesticate horse. Therefore, these variances of the splint bones could be deemed unique to the Dutch Konik and an invaluable tool in the selection for genetic preservation.

## 5. Conclusions 

This study aimed to identify a specific anatomic trait that was non-invasive and unique to the Dutch Konik horse. It focused on comparing the relative lengths of the individual splint bones via post mortem dissections, between the Dutch Konik and domesticated horse. The results from the Dutch Konik limbs (*n* = 47) and modern domesticated horse limbs (*n* = 120), revealed significant differences in relation to the length and symmetry of the 2nd and 4th Metacarpals and Metatarsals. Therefore, displaying an apparent uniqueness in distal limb characteristics indicative to the Dutch Konik horse, which could be an important tool in the selection and preservation of the breed. 

## Figures and Tables

**Figure 1 animals-08-00021-f001:**
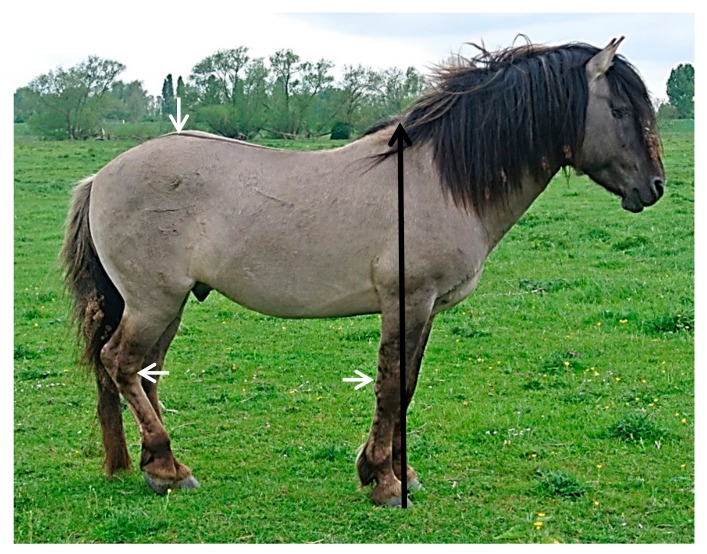
Wither height 130–140 cm (highest Thoracic dorsal spine) represented by the measurement from the ground to the black arrowhead in this Dutch Konik Stallion. Note: primitive stripes (white arrows).

**Figure 2 animals-08-00021-f002:**
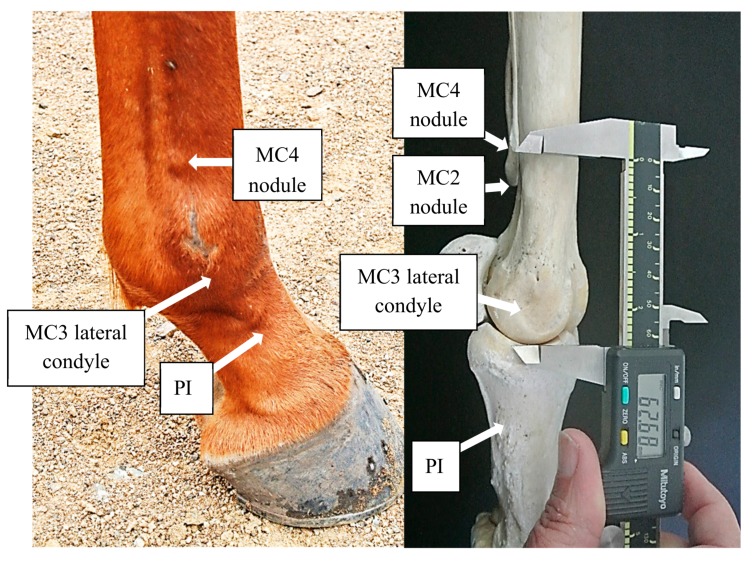
The distal nodule of MC4 is easily visible in the right forelimb of this 5-year-old Thoroughbred (left). The photo on the right demonstrates the technique used for measuring the nodule to condyle distance, in this case of MC4 (mMC4).

**Figure 3 animals-08-00021-f003:**
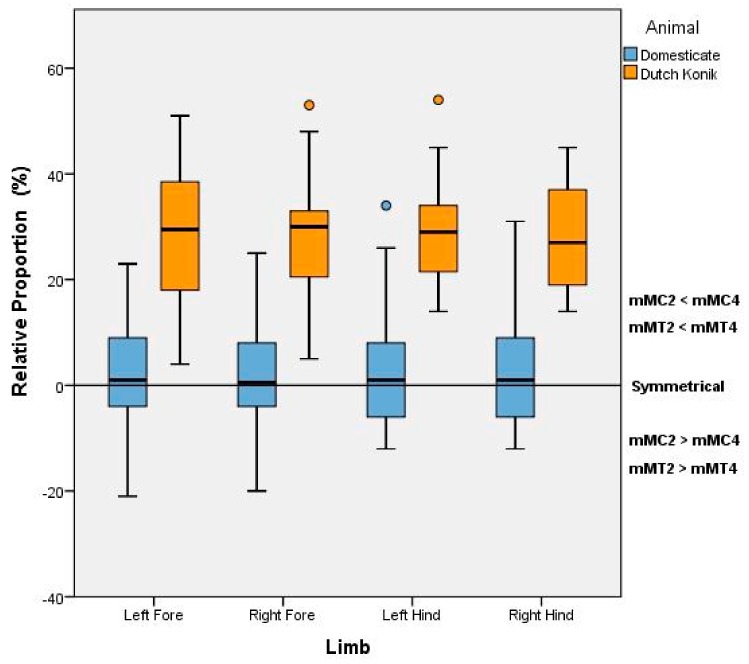
Variation in length and symmetry of the 2nd and 4th Metacarpals (MC) and Metatarsals (MT) between the Dutch Konik and Domesticate Horse, as indicated by their respective nodule to condyle measurements (mMC and mMT).

**Table 1 animals-08-00021-t001:** Measuring techniques for mMC2, mMC4, mMT2 and mMT4.

Descriptor	Measurement Description
mMC2	On the caudomedial aspect, place the fixed caliper arm at the distal point of the nodule on MC2; then extend the movable caliper arm to the distal edge of the medial MC3 condyle.
mMC4	On the caudolateral aspect, place the fixed caliper arm at the distal point of the nodule on MC4; then extend the movable caliper arm to the distal edge of the lateral MC3 condyle.
mMT2	On the caudomedial aspect, place the fixed caliper arm at the distal point of the nodule on MT2; then extend the movable caliper arm to the distal edge of the medial MT3 condyle.
mMT4	On the caudolateral aspect, place the fixed caliper arm at the distal point of the nodule on MT4; then extend the movable caliper arm to the distal edge of the lateral MT3 condyle.

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
