# Peer review of "A Novel Non-Invasive Selection Criterion for the Preservation of Primitive Dutch Konik Horses"

_animals, 2018, doi:10.3390/ani8020021_

Round 1

Reviewer 1 Report

author switches unlogically from Dutch Konik to Polish Konik in the introduction. It was only explained later.

Author Response

Reviewer No. 2

author switches unlogically from Dutch Konik to Polish Konik in the introduction. It was only explained later.

Response: Line 89 replaced with:-

Recent studies of the Dutch Konik’s forefathers in Poland have noted the relative uniformity and variances of certain exterior traits in the distal limb, including the circumference and length of the 3rd Metacarpal (MC3) and 3rd Metatarsal (MT3) [19,20].

Reviewer 2 Report

1.  The breeding selection of the Dutch Konik Horse is different than the breeding selection in domesticated horses, because in domesticated breeds selection is based on abilities which suit the use as working- or sports-horse.

Therefore you could consider the Dutch Konik Horse as a 'primitive' breed. The only factor which troubles this theorie is the fact that there are also abilities which we possibly excluded from the breeding selection of the Dutch Konik Horse: those abilities in favour for prey-animals such as speed or observation skills. Therefore there could have been a selection in favour of the more relaxed, food-intake and energetic efficient type of animal.

2. table 1: text at mMT4: I think 'medial' should be 'lateral'

Author Response

1.  The breeding selection of the Dutch Konik Horse is different than the breeding selection in domesticated horses, because in domesticated breeds selection is based on abilities which suit the use as working- or sports-horse.

Therefore you could consider the Dutch Konik Horse as a 'primitive' breed. The only factor which troubles this theorie is the fact that there are also abilities which we possibly excluded from the breeding selection of the Dutch Konik Horse: those abilities in favour for prey-animals such as speed or observation skills. Therefore there could have been a selection in favour of the more relaxed, food-intake and energetic efficient type of animal.

Response: Noted

2. table 1: text at mMT4: I think 'medial' should be 'lateral'

Response: Corrected

Reviewer 3 Report

Dear Authors, attached please find a pdf with comments.

Regards

Author Response

Alterations have been made to the paper according to the peer reviewer's requests and highlighted in the pdf file attached, except for the following;

Line 33.In this sentence, the reference to the Konik horse was as per by Janikowski, T. 1942 [3].
 May I suggest the following, in a hope that this may suffice?

One such breed is the Konik, a hardy, stocky horse first established in Poland and believed to have descended directly from the now extinct Tarpan (Equus caballus gmelini, Antonius) [2,3].

Line 154. Should 'Domesticate' horse be 'Domesticated' horse? Either will suffice.

Further requests in Line 33 have also been addressed - 1. in line 34 and the other further into the introduction.

Round 2

Reviewer 3 Report

Accept in present form.